# Influence of CO$_2$ Curing on the Alkali-Activated Compound Mineral Admixtures' Corrosion Resistance to NaCl Dry–Wet Alternations

Houchao Sun [1], Wenjie Cheng [2], Haoxin Xu [3], Zhangjie Cai [1], Minggan Yin [1] and Feiting Shi [1,*]

1  School of Civil Engineering, Yancheng Institute of Technology, Yancheng 224051, China
2  School of Mechanical Engineering, Yancheng Institute of Technology, Yancheng 224051, China
3  School of Chemistry & Chemical Engineering, Yancheng Institute of Technology, Yancheng 224051, China
*  Correspondence: shifeiting@ycit.cn

**Abstract:** In this study, the influence of CO$_2$ curing on the corrosion resistance of reinforced alkali-activated compounds is investigated. Fly ash (FA) and blast furnace slag powder (BFS) are used as mineral admixtures. The specimens were subjected to dry–wet alternations with 3% NaCl, used to simulate a concrete structure under a corrosion environment. The ultrasonic velocity, mass loss rate, and electrical characteristics (such as electrical resistance, AC impedance spectra, and corrosion area rates determined by Tafel curves) are utilized to determine the degree of corrosion. Scanning electron microscopy (SEM) and X-ray diffraction (XRD) are used to analyze the corrosion mechanism. Results show that the corrosion resistance is decreased by the addition of FA but improved by CO$_2$ curing. When CO$_2$ curing is provided, the addition of BFS shows a higher enhancing effect on the corrosion resistance than that of FA. The equivalent circuit diagram of reinforced alkali-activated compound mineral admixtures obtained by AC impedance spectra is composed of three electrical elements (electrical resistance and capacitance in parallel) in series. The X-ray diffraction results show that adding BFS and CO$_2$ curing can decrease the rust's iron oxides on the steel bars' surface. Finally, as found in the SEM photos, BFS and CO$_2$ curing can effectively improve the compactness of specimens. Meanwhile, the roughness of hydration is increased by CO$_2$ curing.

**Keywords:** corrosion resistance; CO$_2$ curing; corrosion area rates; NaCl dry–wet alternation; electrical characteristics

## 1. Introduction

Cement-based materials have been broadly used in buildings such as houses, bridges, and highways for many years [1–3]. However, cement production also produces a great number of byproducts [4,5]. To alleviate this issue, mineral admixtures are usually applied to replace cement. Silica fume, fly ash, and slag powder are commonly used additives in cement materials. These solid industrial waste-derived mineral admixtures can increase cement concrete's mechanical strength and durability [6–8]. Meanwhile, cement-based materials with mineral admixtures can also benefit from high-temperature curing to increase their strength and durability [9–11]. Although mineral admixtures are easy to obtain, their incorporation often deteriorates the strength and durability of concrete under standard curing conditions [12,13]. Consequently, alkali-activated mineral admixtures can be an option to replace some cement materials.

Alkali-activated mineral admixtures are composed of mineral admixtures, sodium silicate, alkali, water, etc. [14–17]. It has been pointed out in prior research that this kind of material shows excellent mechanical properties. Cement-based materials with NaOH as an activator show ultra-high mechanical strengths (flexural strength of higher than 20 MPa and compressive strength of higher than 120 MPa) [18–20]. Several studies have been carried out on its mechanical performance, the use of different alkali-activated agents, and

their high-temperature resistance and micro-properties [21]. Little attention has been paid to the influence of NaCl erosion on its performance.

Cementitious materials are always applied with reinforcing steel bars to sustain the load in service [22,23]. Alkali-activated mineral admixtures without steel bars cannot be applied in a real project. Meanwhile, concrete usually encounters chlorine salt erosion when it is used in a coastal environment. As reported in prior research, the alkali-activated cementitious material shows quite small porosity, thus improving the corrosion resistance of steel bars. Moreover, the surface of steel bars' passivation film can be ameliorated by the alkali-activated mineral admixtures, leading eventually to increasing the subsequent corrosion resistance [24,25]. However, NaCl dry–wet alternations usually cause serious erosion to the reinforcement inside cement-based materials [26]. Little attention has been paid to the corrosion resistance of reinforced alkali-activated mineral admixtures under the environment of NaCl dry–wet alternations.

$CO_2$ is the main cause of the greenhouse effect. The production of cement can increase the emissions of $CO_2$, which has been used for curing cement-based materials [27,28]. The reinforced cement matrix's mechanical characteristics and corrosion resistance can be enhanced by $CO_2$ curing, according to earlier studies [29]. However, the effect of $CO_2$ curing on mineral admixtures is still unknown [30–32]. Whether $CO_2$ curing can help improve the corrosion resistance of reinforced alkali-activated mineral admixtures should be investigated. The innovation of this study is to investigate the influence of $CO_2$ curing on the corrosion resistance of reinforced alkali-activated material under the action of NaCl dry–wet alternations. Multiple sets of electrical parameters and ultrasonic velocity, along with X-ray diffraction (XRD) and scanning electron microscopy (SEM), are used for the analysis of corrosion resistance. However, related studies are rarely reported.

In this study, the effect of $CO_2$ curing on mineral admixtures with alkali-activated compounds is investigated. The mass, the ultrasonic velocity, and the electrical resistance of reinforced $CO_2$-cured alkali-activated compound mineral admixtures are determined. Meanwhile, the mass loss rate and the corrosion area rate of steel bars are measured. Finally, the micro-parameters (the scanning electron microscopy (SEM) of the alkali-activated mineral admixtures and X-ray diffraction (XRD) of the rust) are obtained to reveal the corrosion mechanism of the reinforced $CO_2$-cured alkali-activated compound mineral admixtures. The outcomes of this study can provide a guideline for the application of $CO_2$-cured reinforced alkali-activated mineral admixtures, which may be applied in a NaCl erosion environment in the future.

## 2. Experimental Arrangement

### 2.1. Raw Materials

The sodium silicate applied in this research is made by Tongxiang Hengli Chemical Co., Ltd., Tongxiang, China. The sodium silicate shows a melting point of 40 °C~48 °C. S95 blast furnace slag powder (BFS) produced by HSBC New Materials Co., Ltd., Zhengzhou, China, is applied in this research. The BFS shows a specific surface area and an ignition loss of the blast furnace slag powder of 2.88 g/cm³, 435.8 m²/g, and 1.98%, respectively. Secondary fly ash made by the Shijiazhuang Shunli mineral products Co., Ltd., Shijiazhuang City, China, is used in this study. The density of fly ash is 2.4 g/cm³. The sodium hydroxide is manufactured by Shanghai Dongmiao Chemical Technology Co., Ltd., Shanghai, China. Sodium hydroxide with a density of 2.13 g/cm³, a boiling point of 1390 °C, and a purity of 99.9% is used in this research. The density, boiling point, and purity of sodium hydroxide are 2.13 g/cm³, 1390 °C, and 99.9%, respectively.

Tables 1 and 2 display the raw materials' particle passing percentage and chemical composition, respectively. The particle passing percentage of raw materials is measured by negative pressure sieve analysis, according to the Chinese standard GB/T1345-2005 [33]. Meanwhile, the chemical compositions of BFS and FA in Table 2 are determined by X-ray fluorescence spectrometry. All data in Tables 1 and 2 are provided by the manufacturer of raw materials, which are within acceptable limits.

**Table 1.** Particle passing percentage of raw materials (%).

| Types | 0.3 μm | 0.6 μm | 1 μm | 4 μm | 8 μm | 64 μm | 360 μm |
|-------|--------|--------|------|------|------|-------|--------|
| BFS | 0.03 | 0.1 | 3.5 | 19.6 | 35.0 | 97.9 | 100 |
| FA | 12.3 | 41.7 | 66.2 | 100 | 100 | 100 | 100 |

**Table 2.** The chemical composition of cement (%).

| Types | $SiO_2$ | $Al_2O_3$ | $Fe_xO_y$ | MgO | CaO | $SO_3$ | $K_2O$ | $Na_2O$ | $Ti_2O$ | LI |
|-------|---------|-----------|-----------|-----|-----|--------|--------|---------|---------|-----|
| BFS | 34.1 | 14.7 | 0.2 | 9.7 | 35.9 | 0.2 | 3.5 | — | — | — |
| FA | 55.00 | 29.58 | 6.00 | 0.62 | 4.50 | 0.11 | 1.26 | 2.13 | 0.06 | 0.74 |

As shown in Table 1, the particle sizes of BFS average 0.3~64 μm, while the particle sizes of FA average 0.3~1 μm. It can be obtained from Table 1 that FA shows higher fineness than BFS. Moreover, as illustrated in Table 2, FA contains a higher content of $Al_2O_3$ and $Fe_xO_y$, indicating that FA shows higher activity. Additionally, the content of $SiO_2$ in FA is higher than that of BFS; hence, FA is more prone to secondary hydration.

### 2.2. Specimen Preparation

Table 3 lists the mixing proportions of all samples. To prepare the specimen, the sodium hydroxide and potash water glass are mixed using the NJ-160A cement paste mixer at the beginning. The cement paste is mixed at 140 r/min for 2 min and then stirred at 180 r/min for another 2 min. After that, the fresh paste is poured into molds with a size of 50 mm × 50 mm × 50 mm. Each specimen is embedded with a plain round bar showing a diameter of 8 mm and a piece of stainless-steel wire mesh with a 4.75 mm nominal aperture (mesh size). Specimens are stored in a standard curing chamber with a temperature of 20 °C and a relative humidity of 98.6%, and a TH-B concrete carbonization test box, provided by Tianjin Gangyuan test instrument factory, Tianjin, China, is used to provide a $CO_2$ environment with a concentration of 8%, respectively. The modulus of potassium silicate is 1.0. The modulus of potassium silicate means the amount of substance ratio of $SiO_2$:($K_2O$ + NaOH).

**Table 3.** The mixing proportions (kg/m³).

| Specimens | FA | BFS | Alkali Equivalent | W/B | Modulus | $Na_2O \cdot nSiO_2$ | NaOH |
|-----------|-----|-----|-------------------|-----|---------|----------------------|------|
| A1 | 300 | 0 | 3% | 0.3 | 1 | 29.83 | 6.69 |
| A2 | 240 | 60 | 3% | 0.3 | 1 | 29.83 | 6.69 |
| A3 | 180 | 120 | 3% | 0.3 | 1 | 29.83 | 6.69 |
| A4 | 120 | 180 | 3% | 0.3 | 1 | 29.83 | 6.69 |
| A5 | 60 | 240 | 3% | 0.3 | 1 | 29.83 | 6.69 |
| A6 | 0 | 300 | 3% | 0.3 | 1 | 29.83 | 6.69 |

### 2.3. Measurements

2.3.1. Macro-Performance

The JITAI990 ultrasonic detector is used to measure ultrasonic velocity. It can generate sound with a speed range of 1000 m/s to 9999 m/s. The sides of specimens are smooth and flat, and the probes should be stuck on the axis positions with Vaseline. After all these are finished, the signal of ultrasonic velocity is collected. Figure 1 shows the measuring process of the ultrasonic velocity. The detailed measuring process of the ultrasonic velocity can be found in Wang's research [34].

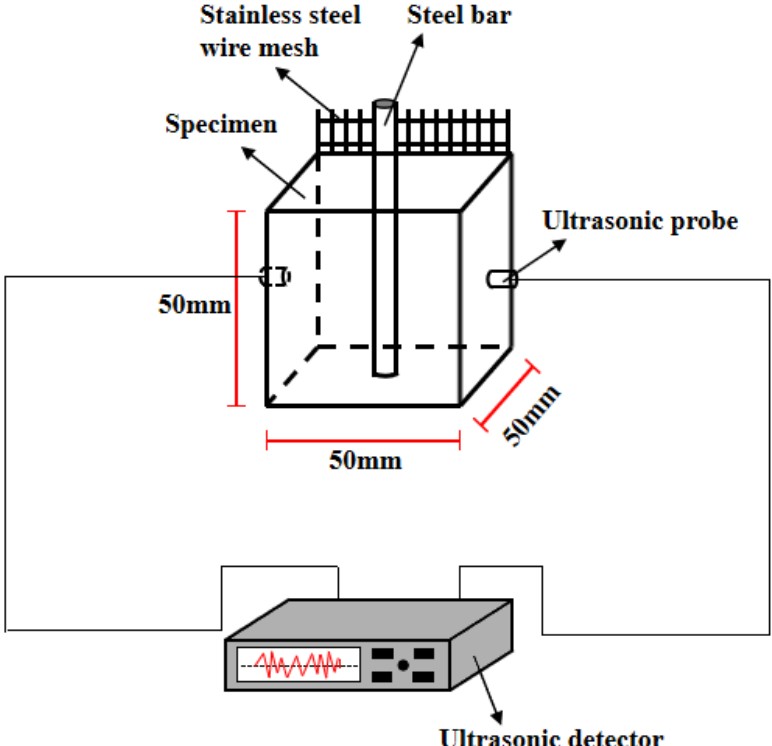

**Figure 1.** The ultrasonic velocity measurement.

The mass loss of the reinforcement can be measured by the following steps.

Firstly, the sanding process is used to remove the original rust on the surface of the steel bars, thus eliminating its effect on the final results. In this study, the original steel bars are polished by a sandblasting machine, and then the impurities on the surface of the steel bars are cleaned with citric acid and moved to be dried and weighed. After that, the standard cured samples with polished steel bars are applied in the corrosion environment. Then, the specimens are destroyed by a press machine, and then the steel bars are taken out and immersed in the citric acid solution for 4 h. Finally, the steel bars are sanded with sandpaper. Additionally, the steel bars are cleaned with water and dried in a vacuum oven at 40 °C before the test. The reinforcement and stainless-steel mesh serve as two electrodes. TH2810D LCR digital bridge with a testing voltage of 1 V and testing frequencies of $10^4$ Hz is used for testing the AC electrical resistance. The Princeton electrochemical workstation with a testing voltage of 10 mV and testing frequencies of 0.01~$10^5$ Hz is used to measure the AC impedance spectrum. The average value of six specimens of each group is the measuring value of each parameter. The Tafel method is applied to measure the corrosion area rate of the reinforced alkali-activated mineral admixtures. The voltage of the Tafel method increases from −0.15 V to 0.15 V. The increasing rate of the voltage is 50 mV/s. The corrosion area rate of the inner reinforcement can be calculated by the following Equation (1).

$$\mu = \frac{Mit}{Fr\rho} \tag{1}$$

where $\mu$ is the corrosion area rate, $M$ is the molar mass of iron, $i$ is the corrosion current density by the Tafel curves. Additionally, $t$, $F$, $r$, and $\rho$ are the corrosion time, Faraday constant, radius of reinforcement, and density of reinforcement. All measuring detail processes can be found in prior research [35,36]. The measurement of AC electrical parameters is shown in Figure 2.

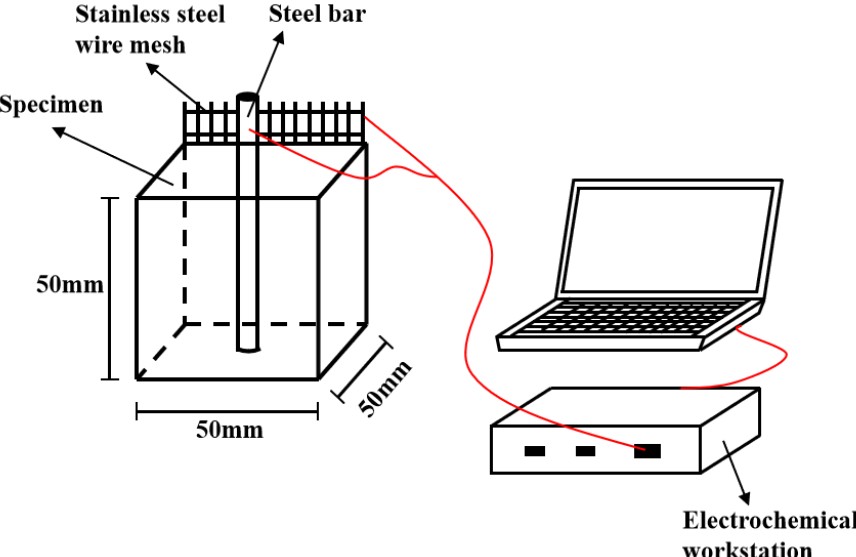

**Figure 2.** Measurement of AC electrical parameters.

2.3.2. Microscopic Performance

Samples the size of a soybean used for SEM measurement are firstly sprayed with gold in a vacuum environment. After that, all samples are moved to the JSM-IT800 high-resolution thermal field emission scanning electron microscope for measurement.

After the experiment of the macro-performance is finished, the reinforced specimens are destroyed. The alkali-activated cementitious matrix on the reinforcement surface is removed, and the rust on the surface of the steel bars is collected. The powdered rust is applied in the determination of X-ray diffraction by the DX-2700BH crystal structure XRD analyzer, provided by Suzhou Shipu Instrument Co., Ltd., Suzhou, China.

## 3. Results and Discussions

### 3.1. The Mass Loss Rate of the Reinforced Alkali-Activated Mineral Admixtures

When the reinforcement is corroded, the rusted reinforcement will expand, thus accelerating the development of cracks in specimens. Therefore, in this study, the mass loss rate of the reinforced alkali-activated mineral admixtures is measured to reflect the corrosion degree. Figure 3 shows the mass loss rate of the reinforced alkali-activated mineral admixtures. As shown in the figure, the mass loss rate is increased by the NaCl D–W alternations and the addition of FA. This can be attributed to the accelerated penetration of chloride ions in materials by NaCl D–W alternations, leading to increasing the corrosion rate of reinforcement [37,38]. More cracks in the specimens are formed by the increased corrosion rate, resulting in higher mass loss rate. Additionally, the mass loss rate is decreased by $CO_2$ curing, which is attributed to the higher content of calcium carbonate formed by the reaction of $CO_2$ and $Ca(OH)_2$. As shown in Figure 3, when FA is used, the decreasing rate of mass loss rate further increases. This is because FA shows a high degree of $[SiO_4]^{4-}$ polymerization in the vitreous structure network, which results in low reaction activity. FA is more difficult to be excited by alkali [39,40]. Therefore, the mass of alkali-activated mineral admixtures is decreased by the addition of FA.

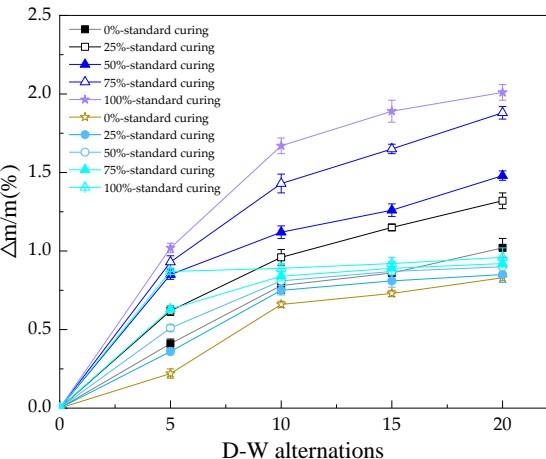

**Figure 3.** Δ m/m of the reinforced alkali-activated mineral admixtures during D–W alternations.

The mass loss rate of the reinforcement is exhibited in Figure 4. It can be observed that the mass loss rate of reinforcement increases with the increasing NaCl D–W alternations and the addition of BFS, due to the increased corrosion degree by D–W alternations and the corrosion resistance by BFS. Meanwhile, $CO_2$ curing demonstrates a positive effect on the corrosion resistance of the reinforcement [41,42]. The result of the reinforcement's mass loss rate is consistent with the mass loss rate of the reinforced alkali-activated mineral admixtures.

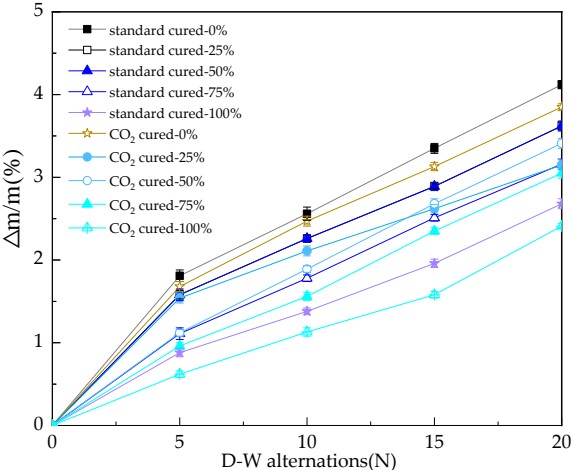

**Figure 4.** The mass loss rate of steel bars during D–W alternations.

## 3.2. The Ultrasonic Velocity of the Reinforced Alkali-Activated Mineral Admixtures

The ultrasonic velocity of the reinforced alkali-activated mineral admixtures is illustrated in Figure 5. It can be seen that the ultrasonic velocity decreases with the increasing NaCl D–W alternations and the decreasing dosage of BFS. This reduction can be attributed to the fact that the NaCl D–W alternations induce the inner cracks of the specimens [43,44]. Additionally, the NaCl D–W alternations accelerate the corrosion of reinforcement, increasing the number and width of cracks, thus decreasing the corresponding ultrasonic velocity. Furthermore, it can be noted that $CO_2$ curing increases the ultrasonic velocity. The more dosages of BFS, the higher the increase. $CO_2$ curing can accelerate the carbonization rate; therefore, the compactness of hydration products is improved, leading eventually to increasing the corresponding ultrasonic velocity. Moreover, the alkali-activated activity of BFS is higher than that of FA. As a consequence, BFS demonstrates a positive effect on the ultrasonic speed.

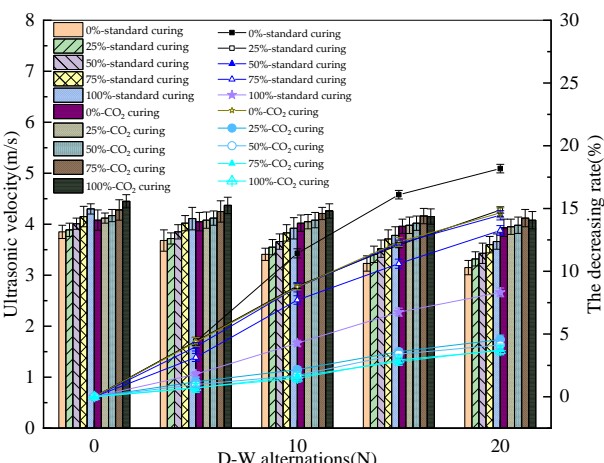

**Figure 5.** The ultrasonic velocity of reinforced alkali-activated mineral admixtures during D–W alternations.

### 3.3. The Electrical Parameters of Reinforced Alkali-Activated Mineral Admixtures

The electrical resistivity of reinforced alkali-activated mineral admixtures is depicted in Figure 6. It can be observed that the electrical resistivity continues to increase with more NaCl D–W alternations. This is because the NaCl D–W alternations can increase the inner cracks of the reinforced alkali-activated mineral admixtures [43,44]. The inner cracks block the electron migration, thus decreasing the electrical conduction and increasing the electrical resistivity. $CO_2$ curing reduces the increasing rate of electrical resistivity due to the improved compactness of the specimens. Additionally, the addition of BFS also increases the electrical resistivity of the reinforced alkali-activated mineral admixtures before D–W alternations. However, as the D–W alternations continue to increase, the increasing rate is decreased by adding BFS and $CO_2$ curing, thus improving the fact that the increasing dosage of BFS is effective in decreasing the corrosion rate of reinforced alkali-activated mineral admixtures and improving the corrosion resistance.

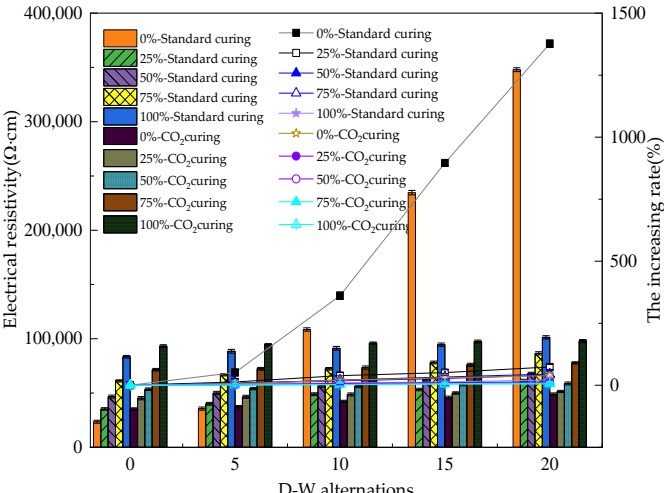

**Figure 6.** The electrical resistivity of reinforced alkali-activated mineral admixtures during D–W alternations.

The Tafel curves of the reinforced alkali-activated mineral admixtures after 20 D–W alternations are shown in Figure 7. It can be seen that the Tafel curves consist of two bifurcation curves. Previous studies reported that the bifurcation point could reflect the corrosion degree of inner reinforcement [35,36]. Additionally, it can be found that the addition of BFS can effectively decrease the potential of the bifurcation point, thus improving the corrosion resistance of concrete under the NaCl dry–wet alternations. This potential can be further reduced with $CO_2$ curing, which validates that $CO_2$ curing is

effective in enhancing corrosion resistance. Table 4 shows the fitted corrosion current obtained using C-view software while measuring Tafel curves. Figure 8 shows the corrosion area rate of the reinforcement calculated by Table 4 and Equation (1).

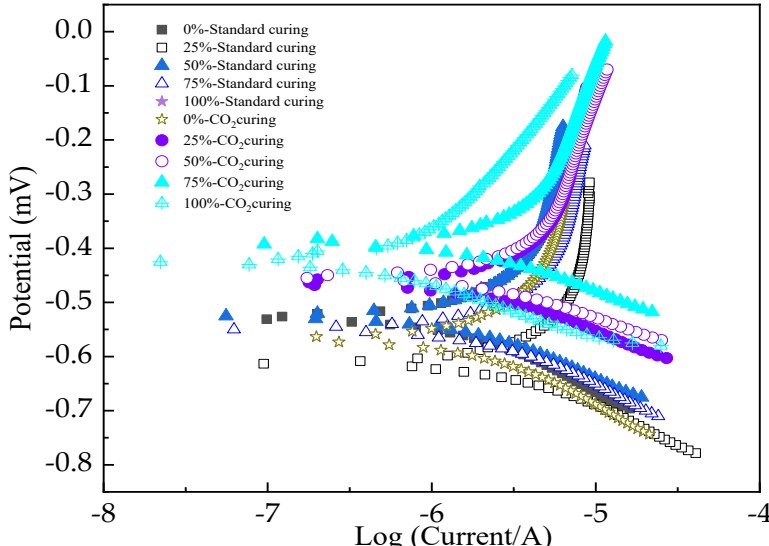

**Figure 7.** The Tafel curves of reinforced alkali-activated mineral admixtures after 20 D–W alternations.

**Table 4.** The corrosion current of reinforced alkali-activated mineral admixtures($10^{-6}$A/cm$^2$).

| Specimens | $A_1$ | $A_2$ | $A_3$ | $A_4$ | $A_5$ |
|---|---|---|---|---|---|
| Standard curing—5 alternations | 3.15 | 2.89 | 2.46 | 2.33 | 0.31 |
| CO$_2$ curing—5 alternations | 3.41 | 3.23 | 2.87 | 2.68 | 0.46 |
| Standard curing—10 alternations | 2.71 | 2.53 | 2.14 | 1.98 | 0.24 |
| CO$_2$ curing—10 alternations | 3.79 | 3.56 | 3.24 | 3.11 | 0.59 |
| Standard curing—20 alternations | 4.12 | 3.98 | 3.66 | 3.27 | 0.89 |
| CO$_2$ curing—20 alternations | 4.08 | 3.79 | 3.60 | 2.96 | 0.67 |

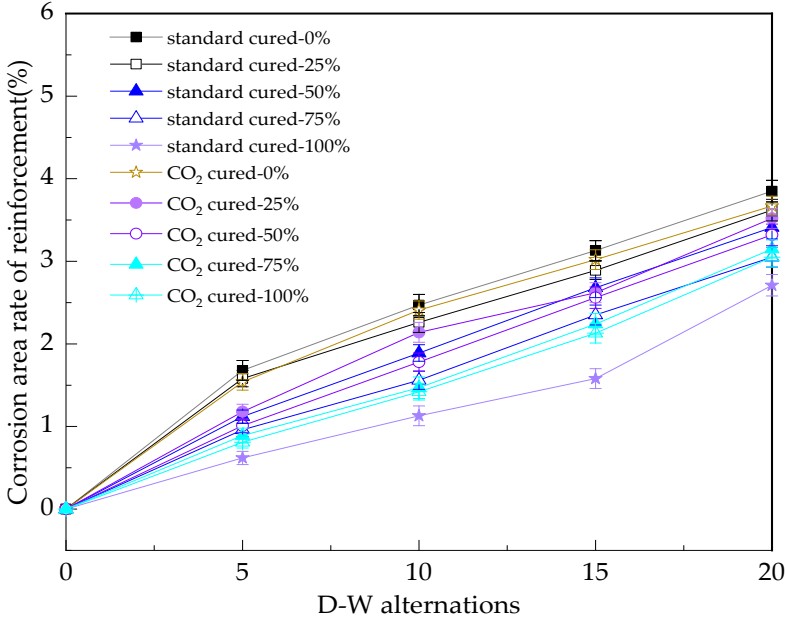

**Figure 8.** The corrosion rate of inner steel bars during D–W alternations.

The corrosion area rate of the reinforcement is shown in Figure 8. This parameter is obtained by testing the Tafel curves. It can be seen that the corrosion area rate of the reinforcement increases with the increasing NaCl D–W alternations and the decreasing dosage of BFS. Additionally, the corrosion area rate decreases with the $CO_2$ curing condition. This is because the NaCl D–W alternations accelerate the corrosion rate of reinforcement, thus increasing the corrosion area rate of the reinforcement. Meanwhile, $CO_2$ curing increases the compactness of the specimens. Hence, the corrosion resistance is improved, and, thus, eventually decreases the corrosion area rate.

In this study, the AC impedance spectrum is obtained to reflect the corrosion degree of the reinforced alkali-activated mineral admixtures. The $Z_i(\Omega)$ and $Z_r(\Omega)$ are the imaginary part and real part, which represent the electrical reactance and resistance of the AC impedance spectrum curves, respectively. Figure 9 demonstrates the AC impedance spectrum of the reinforced alkali-activated mineral admixtures. The minimum point of the AC impedance spectrum increases with the addition of BFS and $CO_2$ curing. This is owing to the fact that FA effectively decreases the electrical resistance of reinforced alkali-activated mineral admixtures, thus decreasing the minimum point of the AC impedance spectrum. In addition, $CO_2$ curing can diminish the number of free ions, leading to increased electrical resistance. Additionally, the minimum point of the AC impedance spectrum increases with the NaCl D–W alternations since the NaCl D–W alternations accelerate the corrosion. The increased rust can block the conductive channel; therefore, the electrical resistance is increased. Moreover, the increasing extent of the minimum point of AC impedance spectrum by NaCl D–W alternations is decreased with the increasing dosage of BFS and the $CO_2$ curing.

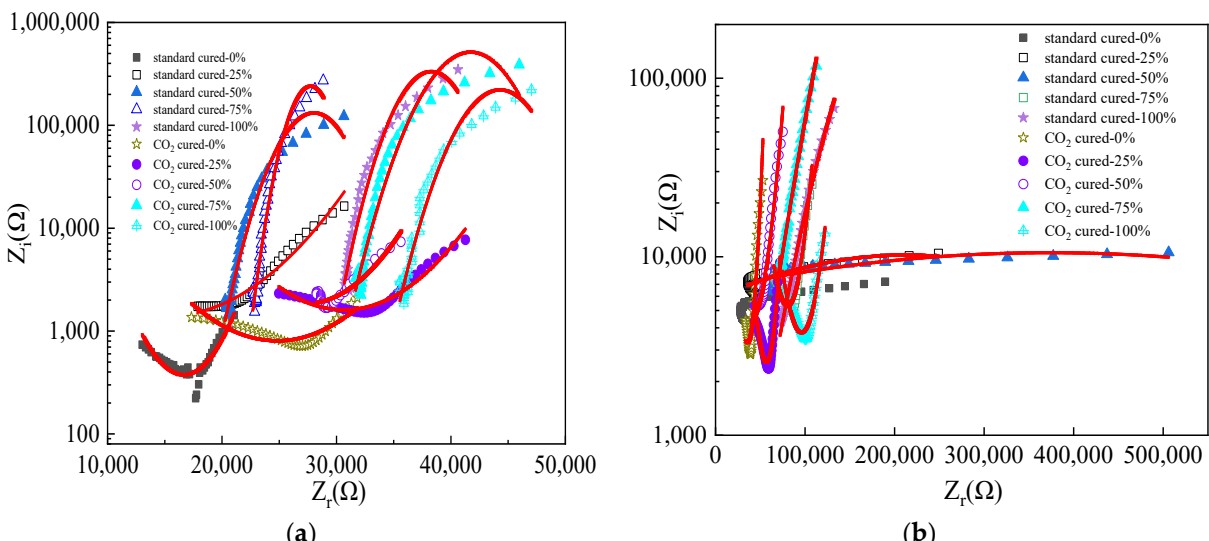

**Figure 9.** The AC impedance spectrum curves of reinforced alkali-activated mineral admixtures during D–W alternations. (**a**) Before NaCl corrosion. (**b**) After 20 NaCl D–W alternations.

All values of the AC impedance spectrum curves were input to ZSimpWin software and the fitting curves of the equivalent circuit diagrams were obtained (the red lines in Figure 9). Additionally, the equivalent circuit diagram of reinforced alkali-activated mineral admixtures is demonstrated in Figure 10. According to the fitting results and the prior studies [44–48], the electric circuit comprises the three electrical resistance and reactance of the rust, the matrix, and the pore solution. As shown in Figure 10, all these electrical elements are in series after a parallel connection (the electrical components of the same substance are in parallel, while the electrical components of different substances are in series). Table 5 shows the chi-square of the fitting curves. As observed in Table 5, the highest chi-squared value of all curves is 0.014, indicating the accuracy of the selected equivalent circuit diagram.

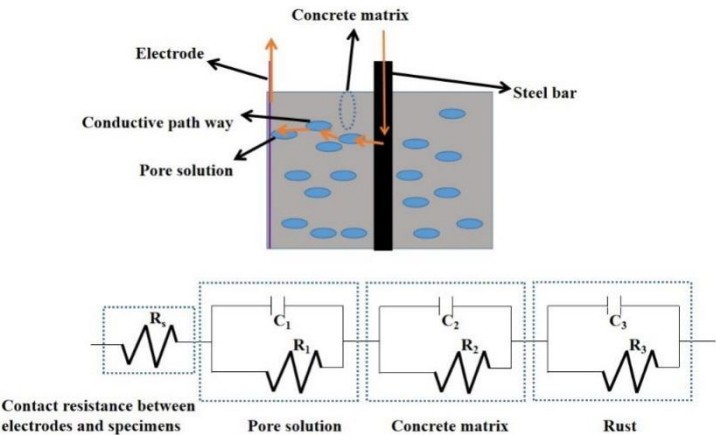

**Figure 10.** Corresponding equivalent circuits of reinforced alkali-activated mineral admixtures.

**Table 5.** The fitting results of the equivalent circuit diagram.

| Specimens | Chi-Square |
|---|---|
| Standard cured-0%-Before NaCl D–W alternation | 0.012 |
| Standard cured-25%-Before NaCl D–W alternation | 0.014 |
| Standard cured-50%-Before NaCl D–W alternation | 0.008 |
| Standard cured-75%-Before NaCl D–W alternation | 0.0056 |
| Standard cured-100%-Before NaCl D–W alternation | 0.013 |
| $CO_2$-cured-0%-Before NaCl D–W alternation | 0.0097 |
| $CO_2$-cured-25%-Before NaCl D–W alternation | 0.012 |
| $CO_2$-cured-50%-Before NaCl D–W alternation | 0.011 |
| $CO_2$-cured-75%-Before NaCl D–W alternation | 0.0089 |
| $CO_2$-cured-100%-Before NaCl D–W alternation | 0.0091 |
| Standard cured-0%-After NaCl D–W alternation | 0.0099 |
| Standard cured-25%-After NaCl D–W alternation | 0.012 |
| Standard cured-50%-After NaCl D–W alternation | 0.01 |
| Standard cured-75%-After NaCl D–W alternation | 0.0087 |
| Standard cured-100%-After NaCl D–W alternation | 0.0065 |
| $CO_2$-cured-0%-After NaCl D–W alternation | 0.0079 |
| $CO_2$-cured-25%-After NaCl D–W alternation | 0.0088 |
| $CO_2$-cured-50%-After NaCl D–W alternation | 0.0092 |
| $CO_2$-cured-75%-After NaCl D–W alternation | 0.0097 |
| $CO_2$-cured-100%-After NaCl D–W alternation | 0.0045 |

The electrical resistivity of the rust of reinforced alkali-activated mineral admixtures is illustrated in Figure 11. As demonstrated in Figure 11, the electrical resistivity of the rust of reinforced alkali-activated mineral admixtures decreases with the increasing dosage of BFS. It can be obtained from Figure 11 that the addition of BFS can help improve the corrosion resistance of reinforced alkali-activated mineral admixtures. Moreover, $CO_2$ curing can decrease the electrical resistance of rust. Therefore, as proved in Figure 11, the corrosion resistance of reinforced alkali-activated mineral admixtures is improved by $CO_2$ curing.

*3.4. The Micro-Structure Research Results*

This research was undertaken to investigate the corrosion resistance of reinforced alkali-activated mineral admixtures. The specimens are alkali-activated FA and alkali-activated BFS, which are cured in the standard curing environment and $CO_2$ curing environment, respectively. The X-ray diffraction (XRD) of the rust on the surface of the reinforcement is shown in Figure 12. Compared with the XRD results in other journals [45–47], the diffraction diagrams of ($\alpha$-FeO(OH) and $\gamma$-Fe(OH)) are observed in the rust, and the height of the diffraction diagrams of $\gamma$-Fe(OH) increases in the order of $CO_2$-cured BFS < $CO_2$-cured FA < standard cured alkali-activated BFS < standard cured

alkali-activated cured FA. This result confirms that adding BFS and $CO_2$ curing effectively improve corrosion resistance against NaCl D–W alternations.

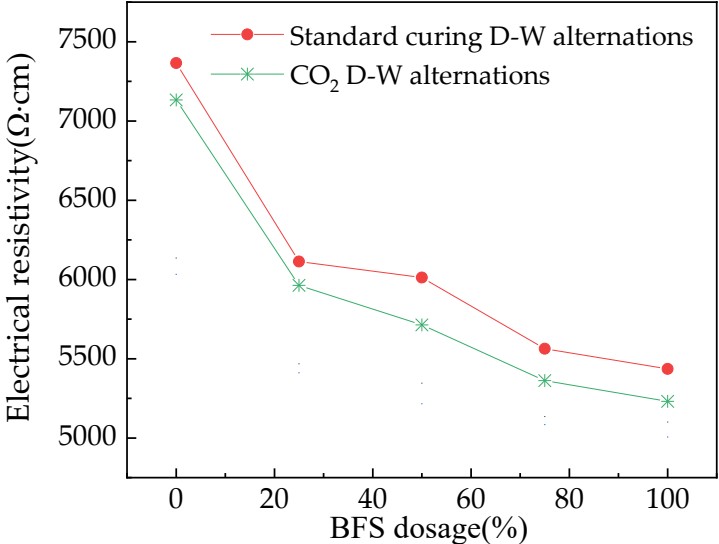

**Figure 11.** The electrical resistivity of the rust through an equivalent circuit diagram.

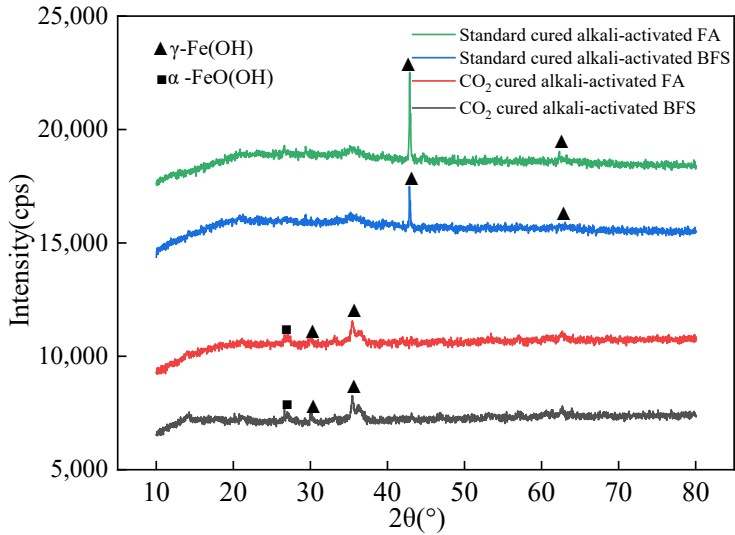

**Figure 12.** The X-ray diffraction (XRD) diagrams of the rust on the surface of steel bars.

The SEM micro-structure photos of standard cured alkali-activated FA, standard cured alkali-activated BFS, and $CO_2$-cured alkali-activated FA specimens are presented in Figure 13. As shown in Figure 13, more needle-like and loose hydration products are found. Additionally, as shown in Figure 13b, the hydration products of standard cured alkali-activated BFS are more compact than those of the standard cured alkali-activated FA, while as shown in Figure 13c, the hydration products of $CO_2$-cured alkali-activated BFS are coarser and more compact. Therefore, the SEM micro-structure photos of specimens further prove that adding BFS and $CO_2$ curing can effectively improve the corrosion resistance of reinforced alkali-activated mineral admixtures exposed to the NaCl D–W alternations.

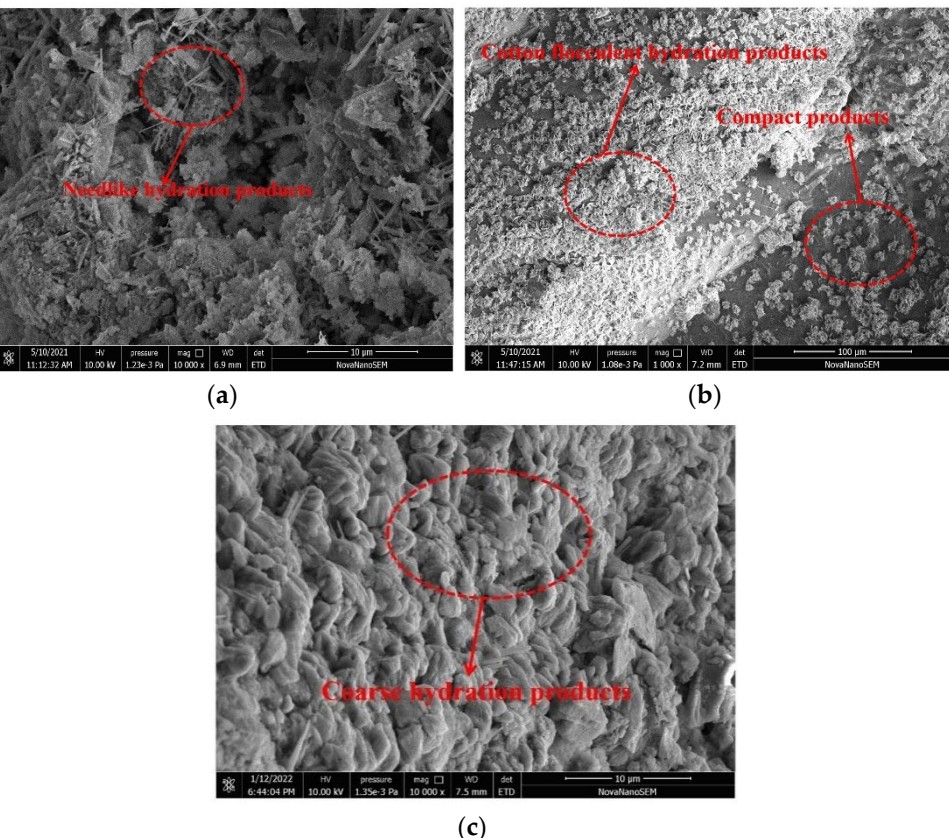

**Figure 13.** SEM micro-structure photos of specimens. (**a**) Standard cured alkali-activated FA. (**b**) Standard cured alkali-activated BFS. (**c**) $CO_2$-cured alkali-activated BFS.

## 4. Conclusions

The mass loss rates of the reinforced alkali-activated mineral admixtures and the inner steel bars increase with the increasing number of NaCl D–W alternations but decrease with the addition of BFS and $CO_2$ curing.

The electrical resistivity of the reinforced alkali-activated mineral admixtures increases with increasing D–W alternations, the addition of BFS, and $CO_2$ curing. Meanwhile, the increasing rate of electrical resistivity by D–W alternations decreases with more dosage of BFS and $CO_2$ curing.

The corrosion area rate of reinforced alkali-activated mineral admixtures and the electrical resistivity of the rust increase with NaCl D–W alternations but decrease with the increasing dosage of blast furnace slag powder and $CO_2$ curing. Adding BFS and $CO_2$ curing can improve the corrosion resistance of reinforced alkali-activated mineral admixtures.

This study will provide an alkali-activated mineral admixture with good corrosion resistance under a chloride corrosion environment.

**Author Contributions:** Conceptualization, F.S. and H.S.; methodology, H.S.; software, W.C.; validation, H.S., H.X. and Z.C.; formal analysis, H.S.; investigation, W.C.; resources, M.Y.; data curation, F.S.; writing—original draft preparation, H.S.; writing—review and editing, F.S.; visualization, H.S.; supervision, F.S.; project administration, F.S.; funding acquisition, H.S. All authors have read and agreed to the published version of the manuscript.

**Funding:** This research was funded by Science and Technology Projects of Ministry of Housing and Urban–Rural Development of China, grant number 2014-K4-024.

**Institutional Review Board Statement:** Not applicable.

**Informed Consent Statement:** Not applicable.



**Data Availability Statement:** The data used to support the findings of this study are available from the corresponding author upon request.

**Acknowledgments:** The authors gratefully acknowledge the financial support of YCIT, and the testing services from Analysis and Test center of YCIT.

**Conflicts of Interest:** The authors declare no conflict of interest.

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
