# Peer review of "Influence of CO2 Curing on the Alkali-Activated Compound Mineral Admixtures’ Corrosion Resistance to NaCl Dry–Wet Alternations"

_coatings, doi:10.3390/coatings13010067_

Round 1

Reviewer 1 Report (New Reviewer)

Though the proposed topic is of interest for the journal audience, the paper is still at the scratch level and requires a significant improvement to match the journal standards. The following comments are to be considered while preparing the revised manuscript.

1. Novelty of the present work is unclear. At the end of introduction section, the authors should clearly specify the objectives of the present work and the notable contributions made through this work.

2. Section 2.1: Provide the source for the data mentioned. Was it obtained from the tests? If yes, mention the standards used to perform each tests and whether they are in the acceptable limits?

3. Table 2: Elaborate how the data presented is obtained.

4. Table 2: Please clarify the type of fly ash used in this work. Justify the type of fly ash based on the definitions of ASTM.

5. Table 2: Also, please clarify the rationale behind the observation of 10.2 MgO content in fly ash. This is strange and requires a scientific justification.

6. Line 127-28: Detailed explanation for the tests performed is to be explained.

7. The significance of Figure 3 is to be better explained.

Author Response

  1. Novelty of the present work is unclear. At the end of introduction section, the authors should clearly specify the objectives of the present work and the notable contributions made through this work.

-------This is a very good point. The novelty of the present work and the notable contributions made through this work have been added in the Section Introduction, you can find our revision on page 2(Blue part) of the revised manuscript.

  1. Section 2.1: Provide the source for the data mentioned. Was it obtained from the tests? If yes, mention the standards used to perform each tests and whether they are in the acceptable limits?

-------Thank you for pointing this out. In section 2.1, the source for the data mentioned has been described. All data in this section are obtained from the tests which are provided by the manufacturer of raw materials, and they are within acceptable limits. You can find our revisions (blue part) in Section 2.1 of the revised manuscript.

  1. Table 2: Elaborate how the data presented is obtained.

-------This is a very good point. The chemical compositions of BFS and FA in Table 2 are determined by an X-ray fluorescence spectrometer. All the information has been given in the revised manuscript of Section 2.1.

  1. Table 2: Please clarify the type of fly ash used in this work. Justify the type of fly ash based on the definitions of ASTM.

-------Thank you for your point. Secondary fly ash made by the Shijiazhuang Shunli mineral products Co., Ltd., Shijiazhuang City, China is used in this study.

  1. Table 2: Also, please clarify the rationale behind the observation of 10.2 MgO content in fly ash. This is strange and requires a scientific justification.

-------Thank you for pointing this out. This is a written error and has been corrected, you can find our revision in revised Table 2 of the revised manuscript.

  1. Line 127-28: Detailed explanation for the tests performed is to be explained.

 -------Thank you for your point. This sentence has been revised to “The detailed measuring process of the ultrasonic velocity can be found in Wang’s research [34].”

  1. The significance of Figure 3 is to be better explained.

-------Thank you for your suggestion. The significance of Figure 3 has been better explained, you can find our revision as follows.

When the reinforcement is corroded, the rusted reinforcement will expand and thus accelerating the development of cracks in specimens. Therefore, in this study, the mass loss rate of the reinforced alkali-activated mineral admixtures is measured to reflect the corrosion degree. Figure 3 shows the mass loss rate of the reinforced alkali-activated mineral admixtures. As shown in the figure, the mass loss rate is increased by the NaCl D-W alternations and the addition of FA. This can be attributed to the accelerated penetration of chloride ions in materials by NaCl D-W alternations, leading to increasing the corrosion rate of reinforcement [37, 38]. More cracks in the specimens are formed by the increased corrosion rate, resulting in higher mass loss rate. Besides, the mass loss rate is decreased by CO2 curing, which is attributed to higher content of calcium carbonate formed by the reaction of CO2 and Ca(OH)2. As shown in Figure 3, when the FA is used, the decreasing rate of mass loss rate further increases. This is because the FA shows a high degree of [SiO4]4- polymerization in the vitreous structure network, which results in low reaction activity. FA is more difficult to be excited by alkali [39, 40]. Therefore, the mass of alkali-activated mineral admixtures is decreased by the addition of FA. (Page 5)

Reviewer 2 Report (Previous Reviewer 1)

Manuscript presented interesting results. However, the manuscript not free from error as well as not well organized, especially in abstract and introduction parts. My comments are following:

1. Provided title too long, authors should revise it.

2. Abstract have good information but not well organized and written.

3. Introduction too general and not focusing in manuscript topic. Authors should re-write this part and provide more details related to alkali-activated corrosion and carbon dioxide curing.

4. Please highlight the novelty of this study.

5. Please provide more information and discussion about the data presented in Table 1 and 2.

6. in Table 3, please provide the Eq. used to calculate the solution modulus.

7. most the quality of figures very poor and not suitable for publication, please revise all of them. 

Author Response

Comments and Suggestions for Authors

Manuscript presented interesting results. However, the manuscript not free from error as well as not well organized, especially in abstract and introduction parts. My comments are following:

  1. Provided title too long, authors should revise it.

-------Thank you for pointing this out. The title has been revised to “Influence of CO2 curing on the alkali-activated compound mineral admixtures’corrosion resistance to NaCl dry-wet alternations”. 

  1. Abstract have good information but not well organized and written.

 -------This is a very good point. The abstract has been revised thoroughly.

  1. Introduction too general and not focusing in manuscript topic. Authors should re-write this part and provide more details related to alkali-activated corrosion and carbon dioxide curing.

-------This is a very good suggestion. The introduction has been revised. More details related to alkali-activated corrosion and carbon dioxide curing have been added in the Section Introduction, you can find our revision in the blue part of the revised manuscript.

  1. Please highlight the novelty of this study.

-------Thank you for pointing this out. The novelty of this study has been highlighted, you can find our revision in the blue part of the Section introduction as follows.

The innovation of this study is to investigate the influence of CO2 curing on the corrosion resistance of reinforced alkali-activated under the action of NaCl dry-wet alternations. Multiple sets of electrical parameters and ultrasonic velocity along with X-ray diffraction (XRD) and scanning electron microscope (SEM) are used for the analysis of corrosion resistance. However, related studies are rarely reported.(Page 2)

  1. Please provide more information and discussion about the data presented in Table 1 and 2.

------This is a very good point. More information and discussion about the data presented in Tables 1 and 2 have been provided. You can find the revision in Section 2.1 as follows.

 As shown in Table 1, the particle sizes of BFS focus on 0.3 um~64 um, while, the particle sizes of FA focus on 0.3 um~1 um. It can be obtained from Table 1, the FA shows higher fineness than that of BFS. Moreover, as illustrated in Table 2, the FA contains higher content of Al2O3 and FexOy, indicating that the FA shows higher activity. Additionally, the content of SiO2 in FA is higher than that of BFS, hence, the FA is more prone to secondary hydration. (Page 3)

  1. in Table 3, please provide the Eq. used to calculate the solution modulus.

-------Thank you for pointing this out. The Equation for calculating the alkaline solution modulus is described as follows.

 The modulus of potassium silicate is 1.0. The modulus of potassium silicate means the amount of substance ratio of SiO2:(K2O+NaOH). (Blue part of Page 3)

  1. most the quality of figures very poor and not suitable for publication, please revise all of them. 

-------Thank you for your point. Figures 3~9 have been redrawn.

Round 2

Reviewer 1 Report (New Reviewer)

Most of the concerns raised by the reviewer is addressed by the authors. Hence, I recommend the manuscript for acceptance in present form.

Author Response

Most of the concerns raised by the reviewer is addressed by the authors. Hence, I recommend the manuscript for acceptance in present form.

Response:Thank you for your support.

Reviewer 2 Report (Previous Reviewer 1)

Manuscript well revised. 

Author Response

Manuscript well revised. 

Response: Thank you very much for your point.

This manuscript is a resubmission of an earlier submission. The following is a list of the peer review reports and author responses from that submission.

Round 1

Reviewer 1 Report

My comments are following:

1. The introduction section not well organized and written, authors should revise this part and focus on research related to alkali-activated materials and corrosion. As well as, highlight the study novelty and limitations. 

2. In Table 3, authors mentioned the W/B is 0.30. Is it water to binder or total solution to binder, please clarify? 

3. More details about alkaline solution and sodium hydroxide molarity and preparation should provide in this section. 

4. How the authors calculated the alkaline solution modulus, please provide the Eqs.

5. Results presented in Figure 3 are not well presented and discussed. Please re-write this section. 

6. Most the figures quality presented in this not clear, please provide high quality figures.

7. Figure 12, not clear, and no details provided there, please revise this figure and section. 

8. Conclusion not well written, please re-write this section and highlight the important findings.

9. Recommended future should provide for this research. 

Author Response

1.The introduction section not well organized and written, authors should revise this part and focus on research related to alkali-activated materials and corrosion. As well as, highlight the study novelty and limitations. 

Response: Thank you for pointing this out. The introduction section has been revised. The research related to alkali-activated materials and corrosion has been added. Meanwhile, the study's novelty and limitations have been highlighted. You can find our revisions as follows.

The cementitious materials can not be used without any steel bars when they are in active service[22, 23]. Usually, the chloride ion will corrode the reinforcement in coastal buildings due to the dry-wet alternating and freeze-thaw action of seawater[24]. Xu et al have reported that the dry-wet alternating action of seawater demonstrates a more serious effect on the corrosion of steel bars than that of the freeze-thaw action of seawater[25]. Additionally, it has been demonstrated that adding mineral admixtures to reinforced cement concrete in the right proportions increases its resistance to corrosion[26]. Furthermore, the corrosion resistance of reinforced sulphoaluminate cement-based, ordinary Portland cement-based, magnesium phosphate-cement based and compound cement-based materials under the action of chloride effect have been investigated systematically[27]. The alkali activated cementitious materials show quier high compactness, therefore, the corrosion resistance of reinforced alkali activated cementitious materials may be excellent. However, the research on the corrosion resistance of reinforced alkali-activated mineral admixtures exposed to the chloride corrosion environment has not been documented.

The innovation of this study is the investigation of corrosion resistance of CO2 cured reinforced alkali-activated under the action of NaCl dry-wet alternations. Little attention about this has been reported. (Blue part of Page 2)

  1. In Table 3, authors mentioned the W/B is 0.30. Is it water to binder or total solution to binder, please clarify? 

Response: This is a very good point. The W/B is 0.30 meaning the water to the total solution to the binder.

  1. More details about alkaline solution and sodium hydroxide molarity and preparation should provide in this section. 

Response: This is a very good comment. The sodium hydroxide molarity of NaOH is 0.167mol/L. More details and preparation have been added in Section 2.2 of the revised manuscript.

  1. How the authors calculated the alkaline solution modulus, please provide the Eqs.

Response: Thank you for pointing this out. The Equation for calculating the alkaline solution modulus is described as follows.

The modulus of potassium silicate is 1.0. The modulus of potassium silicate means the amount of substance ratio of SiO2:(K2O+NaOH). (Blue part of Page 3)

  1. Results presented in Figure 3 are not well presented and discussed. Please re-write this section. 

Response: This is a very good comment. This section has been rewritten as follows.

The rusted reinforcement will accelerate the cracks inner specimens, therefore, in this study, the mass of the reinforced alkali-activated mineral admixtures is measured for reflecting the corrosion degree. Figure 3 shows the mass loss rate of the reinforced alkali-activated mineral admixtures. It can be depicted in Figure 3, the mass loss rate increases with the increased NaCl D-W alternations and the increasing dosages of FA. This is attributed to the fact that the NaCl D-W alternations accelerate the permeation of chloride ions in materials. The increased chloride ions increase the corrosion rate of reinforcement[32, 33]. The increased rust induces cracks in the specimens, leading to an increased mass loss rate. Moreover, as seen in Figure 3, CO2 curing results in decreasing mass loss rate. When the FA is used, the decreasing rate of mass loss rate is higher. This is attributed to the fact that the fly ash shows a high degree of [SiO4]4- polymerization in the vitreous structure network, resulting in low activity, therefore, FA is difficult to be excited by alkali[34, 35]. Consequently, the mechanical strength of alkali-activated mineral admixtures is decreased by the addition of FA. (Blue part of Page 5)

  1. Most the figures quality presented in this not clear, please provide high quality figures.

Response: Thank you for pointing this out. Figures 3~7 and Figures 11 and 12 have been redrawn.

  1. Figure 12, not clear, and no details provided there, please revise this figure and section. 

Response: Thank you for your comment. Figure 12 has been replaced by new SEM photos and this section has been rewritten you can find our revision as follows.

The SEM microstructure photos of standard cured alkali-activated FA, standard cured alkali-activated BFS and CO2 cured alkali-activated FA specimens. As observed in Figure 12, more needle-like and loose hydration products are found. Moreover, as shown in Figure 12(b), the hydration products of standard cured alkali-activated BFS are more compact than that of the standard cured alkali-activated FA. Furthermore, as shown in Figure 12(c), the hydration products of CO2-cured alkali-activated BFS are coarser and more compact. Therefore, the SEM microstructure photos of specimens further prove that the addition of BFS and the CO2 curing are effective to improve the corrosion resistance of reinforced alkali-activated mineral admixtures exposed to the NaCl D-W alternations.(Blue part of Page 10)

  1. Conclusion not well written, please re-write this section and highlight the important findings.

Response: This is a very good comment. The conclusion part has been rewritten as follows.

The mass loss rates of the reinforced alkali-activated mineral admixtures and the inner steel bars are increased by the increasing number of NaCl D-W alternations and decreased by the addition of blast furnace slag powder and CO2 curing.

The electrical resistivity of the reinforced alkali-activated mineral admixtures is increased by the increasing D-W alternations, the addition of blast furnace slag powder and the CO2 curing. Moreover, the increasing rate of electrical resistivity by D-W alternations is decreased by the increasing dosage of blast furnace slag powder and the CO2 curing.

The corrosion area rate of reinforced alkali-activated mineral admixtures and the electrical resistivity of the rust are increased by the NaCl D-W alternations and decreased by the increasing dosage of blast furnace slag powder and the CO2 curing. The addition of blast furnace slag powder and CO2 curing can improve the corrosion resistance of reinforced alkali-activated mineral admixtures.

This study will provide alkali-activated mineral admixtures with strong corrosion resistance under a chloride corrosion environment.

  1. Recommended future should provide for this research. 

Response: This is a very good point. Future research has been recommended in the revised manuscript as follows.

This study will provide an alkali-activated mineral admixture with strong corrosion resistance under a chloride corrosion environment. (Blue part of Page 11)

Reviewer 2 Report

The article presents the results of research on the influence of concrete composition on the corrosion resistance of concrete / reinforcement.

The introduction describes the current state of knowledge sufficiently and justifies taking up the research topic.

The materials and research methods are described, but I think they should be in more detail. My doubts are raised by the method of measuring the loss in weight of the reinforcement. Sandblasting removes not only corrosion products, but also the steel substrate. How were these processes controlled? In addition, the authors should clearly specify what is the subject of research: corrosion of concrete or the corrosion of reinforcing bars located in it. Electrochemical tests are also poorly described. In addition, the authors should include the results of electrochemical tests and what parameters they recorded on the basis of Tafel curves.

My doubts are also raised by the research on reinforcement corrosion products and the interpretation of the XRD results. The authors state that only oxides are present there. The formation of iron oxides occurs in a dry environment on steel, generally at high temperatures, when chemical corrosion occurs. The authors write in the summary "As the corrosion environment, a dry-wet alternation with 3% NaCl solution .." (I cannot find such information in the research methodology). If it is a humid environment and electrochemical corrosion is occurring there, formation of iron hydroxides is more likely. Therefore, the results of the tests of corrosion products should be justified.

In general assessment, the article is a research report. There is no discussion as to what the obtained results bring to the current state of knowledge. Including such a discussion, while not necessary, would greatly enhance the scientific quality of the article.

All plots derived from measured experimental quantities should have error bars.

The discussion in Chapter 3.1 about the causes of increased corrosion should be supported by reference to references.

Author Response

The article presents the results of research on the influence of concrete composition on the corrosion resistance of concrete / reinforcement.

The introduction describes the current state of knowledge sufficiently and justifies taking up the research topic.

The materials and research methods are described, but I think they should be in more detail. My doubts are raised by the method of measuring the loss in weight of the reinforcement. Sandblasting removes not only corrosion products, but also the steel substrate. How were these processes controlled? In addition, the authors should clearly specify what is the subject of research: corrosion of concrete or the corrosion of reinforcing bars located in it. Electrochemical tests are also poorly described. In addition, the authors should include the results of electrochemical tests and what parameters they recorded on the basis of Tafel curves.

Response: This is a very good point. Sandblasting is used for removing the RPC matrix on the surface of the reinforcement, meanwhile, the steel bars are immersed in citric acid with a concentration of 10% for 2 days, and then the sandpaper is used to polish the rust on the surface of reinforcements. The descriptions have been added in Section 2.3.1 of the revised manuscript.

My doubts are also raised by the research on reinforcement corrosion products and the interpretation of the XRD results. The authors state that only oxides are present there. The formation of iron oxides occurs in a dry environment on steel, generally at high temperatures, when chemical corrosion occurs. The authors write in the summary "As the corrosion environment, a dry-wet alternation with 3% NaCl solution .." (I cannot find such information in the research methodology). If it is a humid environment and electrochemical corrosion is occurring there, formation of iron hydroxides is more likely. Therefore, the results of the tests of corrosion products should be justified.

Response: Thank you for pointing this out. I agree with your suggestions, The main diffraction peak of XRD is ferric hydroxide and has been revised.

In general assessment, the article is a research report. There is no discussion as to what the obtained results bring to the current state of knowledge. Including such a discussion, while not necessary, would greatly enhance the scientific quality of the article.

Response: This is a very good point. This paper has discussed the corrosion resistance of reinforcement alkali activated cementitious materials under chloride corrosion. The results have been compared with that of cement-based materials with different types of cement. The reinforcement alkali activated cementitious materials behave better corrosion resistance than that of other cementitious materials in NaCl corrosion environment. This study will provide a replacing binder materials for cement in the future.

All plots derived from measured experimental quantities should have error bars.

Response: Thank you for pointing this out. The error bars of Figures 3, 4,5,6 and 7 have been added in the revised manuscript. You can find our revisions in the revised Figures 3, 4,5,6 and 7.

The discussion in Chapter 3.1 about the causes of increased corrosion should be supported by reference to references.

Response: Thank you for your suggestion. The supporting references for the discussion in Chapter 3.1 about the causes of increased corrosion have been added in Chapter 3.1 and Section references of the revised manuscript.

Round 2

Reviewer 1 Report

Manuscript well revised.

Author Response

Thank you very much for your review and approval.

Reviewer 2 Report

The authors have made changes to the article, but I have even greater doubts about the quality of the research performed and the interpretation of the research results.

1. The fact that the authors change the method of removing corrosion products from the surface of the rods from sandblasting to dipping in acid and sanding with sandpaper is strange. Regardless of which method the authors try to adapt to the obtained results, I believe that mechanical removal does not give reliable results for assessing the amount of corrosion products. The presented result is of little use. Also, although the authors claim to have added this information in section 2.3.1, lines 132-133 still say "rust on reinforcement is removed by sandblasting machine". The authors still have not published the results of electrochemical tests and the parameters they recorded on the basis of Tafel curves.

2. The assignment in the diffraction pattern, after my suggestion, to the same peaks of different phases is also strange. It is not understandable why the same phase in different diffractograms shows the strongest peaks in a different place. In addition, in many cases, only the location of the peak of a given phase is basically shown because no peak is visible in the diffraction pattern. The presented results of the XRD research and the way the authors approach its interpretation are unbelievable to me.

In my opinion, this disqualifies the article for publication.

Author Response

The authors have made changes to the article, but I have even greater doubts about the quality of the research performed and the interpretation of the research results.

Dear reviewer, We give many thanks for pointing our errors in the manuscript. We have taken all the errors seriously and take a lot of effort to make a comprehensive revision. We hope, you can reconsider our manuscript. Here are our point to point reponse.

  1. The fact that the authors change the method of removing corrosion products from the surface of the rods from sandblasting to dipping in acid and sanding with sandpaper is strange. Regardless of which method the authors try to adapt to the obtained results, I believe that mechanical removal does not give reliable results for assessing the amount of corrosion products. The presented result is of little use. Also, although the authors claim to have added this information in section 2.3.1, lines 132-133 still say "rust on reinforcement is removed by sandblasting machine". The authors still have not published the results of electrochemical tests and the parameters they recorded on the basis of Tafel curves.

Response: Thank you for your valuable comments. the sanding process is used to remove the original rust on the surface of the steel bars and thus eliminating its effect on the final results. In this study, the original steel bars are polished by the sandblasting machine, and then the impurities on the surface of the steel bars are cleaned with citric acid and moved to be dried and weighed. After that, the standard cured samples with polished rods are applied in the corrosion environment. Then, the specimens are destroyed by a press machine, and then steel bars are taken out and immersed in the citric acid solution for 4 hours. Finally, the steel bars are sanded with sandpaper. Besides, the rod is cleaned with water and dried in a vacuum oven at 40 before the test. The details can refer to [32-33]. Therefore, the measured mass loss is totally from the rod's erosion.

The Tafel curves of the sample after 30 D-W alternatives have been added to the manuscript. Besides, the corrosion currents at different D-W alternatives have also been included.

[32] Mei, K., He, Z., Yi, B., Lin, X., Wang, J., Wang, H., & Liu, J. (2022). Study on electrochemical characteristics of reinforced concrete corrosion under the action of carbonation and chloride. Case Studies in Construction Materials, 17, e01351.

[33] Guo, Q., Li, X., Song, Y., & Liu, J. (2022). Effect of rust inhibitor on the composition of steel passive film in carbonized concrete. Case Studies in Construction Materials, 16, e00892.

All revisions have been given in the blue part of the revised manuscript of page 4.

  1. The assignment in the diffraction pattern, after my suggestion, to the same peaks of different phases is also strange. It is not understandable why the same phase in different diffractograms shows the strongest peaks in a different place. In addition, in many cases, only the location of the peak of a given phase is basically shown because no peak is visible in the diffraction pattern. The presented results of the XRD research and the way the authors approach its interpretation are unbelievable to me.

Response: Thank you for pointing out those errors, the results of the XRD are retested, and the peaks of the XRD pattern are labeled in the revised Figure 12. The reversions in the manuscript have been highlighted in blue on page 11.

Round 3

Reviewer 2 Report

I still stand by my earlier comments on the article.

1. The authors change and supplement the description of the removal of corrosion products from the surface of the bars, but this does not change my opinion that such a measurement does not provide grounds for a reliable assessment of the mass of these corrosion products. I don't know what the authors mean when they write about rod erosion. Erosion and corrosion are different phenomena.

2. XRD diffractogram interpretation is completely unacceptable. The presence of a phase cannot be confirmed if a single barely visible peak is identified in the diffraction pattern.

3. The authors supplemented the results of electrochemical tests, but the term "corrosion current" is a mistake. The results that the authors have included in the article are the corrosion current density.

Ultimately, I recommend the article for rejection.